# Bayesian Structure Learning with Generative Flow Networks

**Tristan Deleu**[1]     **António Góis**[1]     **Chris Emezue**[2,*]     **Mansi Rankawat**[1]

**Simon Lacoste-Julien**[1,4]     **Stefan Bauer**[3,5]     **Yoshua Bengio**[1,4,6]

[1]Mila, Université de Montréal   [2]Technical University of Munich   [3]KTH Stockholm

[4]CIFAR AI Chair   [5]CIFAR Azrieli Global Scholar   [6]CIFAR Senior Fellow

## Abstract

In Bayesian structure learning, we are interested in inferring a distribution over the directed acyclic graph (DAG) structure of Bayesian networks, from data. Defining such a distribution is very challenging, due to the combinatorially large sample space, and approximations based on MCMC are often required. Recently, a novel class of probabilistic models, called Generative Flow Networks (GFlowNets), have been introduced as a general framework for generative modeling of discrete and composite objects, such as graphs. In this work, we propose to use a GFlowNet as an alternative to MCMC for approximating the posterior distribution over the structure of Bayesian networks, given a dataset of observations. Generating a sample DAG from this approximate distribution is viewed as a sequential decision problem, where the graph is constructed one edge at a time, based on learned transition probabilities. Through evaluation on both simulated and real data, we show that our approach, called DAG-GFlowNet, provides an accurate approximation of the posterior over DAGs, and it compares favorably against other methods based on MCMC or variational inference.

## 1 INTRODUCTION

Bayesian networks (Pearl, 1988) are a popular framework of choice for representing uncertainty about the world in multiple scientific domains, including medical diagnosis (Lauritzen and Spiegelhalter, 1988; Heckerman and Nathwani, 1992), molecular biology (Friedman, 2004; Sebastiani et al., 2005), and ecological modeling (Varis and Kuikka, 1999; Marcot et al., 2006). For many applications, the structure

of the Bayesian network, represented as a directed acyclic graph (DAG) and encoding the statistical dependencies between the variables of interest, is assumed to be known based on knowledge from domain experts. However, when this graph is unknown, we can learn the DAG structure of the Bayesian network from data alone in order to discover these statistical (or possibly causal) relationships. This may form the basis of novel scientific theories.

Given a dataset of observations, most of the existing algorithms for structure learning return a single DAG (or a single equivalence class; Chickering, 2002), and in practice those may lead to poorly calibrated predictions (Madigan et al., 1994), especially in cases where data is limited. Instead of learning a single graph candidate, we can view the problem of structure learning from a Bayesian perspective and infer the posterior over graphs $P(G \mid \mathcal{D})$, given a dataset of observations $\mathcal{D}$, to account for the epistemic uncertainty over models. Except in limited settings (Koivisto, 2006; Meilă and Jaakkola, 2006), characterizing a whole distribution over DAGs remains intractable because of its combinatorially large sample space and the complex acyclicity constraint. Therefore, we must often resort to approximations of this posterior distribution, e.g., based on MCMC or, more recently, variational inference.

In this paper, we propose to use a novel class of probabilistic models called *Generative Flow Networks* (GFlowNets; Bengio et al., 2021a,b) to approximate this posterior distribution over DAGs. A GFlowNet is a generative model over discrete and composite objects that treats the generation of a sample as a sequential decision problem. This makes it particularly appealing for modeling a distribution over graphs, where sample graphs are constructed sequentially, starting from the empty graph, by adding one edge at a time. In the context of Bayesian structure learning, we also introduce improvements over the original GFlowNet framework, including a novel flow-matching condition and corresponding loss function, a hierarchical probabilistic model for forward transitions, and using additional tools from the literature on Reinforcement Learning (RL). We call our method *DAG-*

---

Correspondence to: Tristan Deleu <deleutri@mila.quebec>

*Work done during an internship at Mila

*Accepted for the 38th Conference on Uncertainty in Artificial Intelligence* (UAI 2022).

*GFlowNet*, to emphasize that the support of the distribution induced by the GFlowNet is exactly the space of DAGs, unlike some variational approaches that may sample cyclic graphs (Annadani et al., 2021; Lorch et al., 2021). Compared to MCMC, which operates through local moves in the sample space (here, adding or removing edges of a graph) and is therefore subject to slow mixing (Friedman and Koller, 2003), DAG-GFlowNet yields a sampling process that samples iid. DAGs, each of them constructed from scratch.

We evaluate DAG-GFlowNet on various problems with simulated and real data, on both discrete and linear-Gaussian Bayesian networks. Furthermore, we show that DAG-GFlowNet can be applied on both observational and interventional data, by modifying standard Bayesian scores (Cooper and Yoo, 1999). On smaller graphs, we also show that it is capable of learning an accurate approximation of the exact posterior distribution. The code is available online.[1]

## 2   RELATED WORK

**Markov chain Monte Carlo**   Methods based on MCMC have been particularly popular in Bayesian structure learning to approximate the posterior distribution. Structure MCMC (MC³; Madigan et al., 1995) simulates a Markov chain in the space of DAGs, through local moves (e.g. adding or removing an edge). Working directly with DAGs leads to slow mixing though; to improve mixing, Friedman and Koller (2003) proposed a sampler in the space of node orders, that introduced a bias (Ellis and Wong, 2008). This was further refined by either modifying the underlying space of the Markov chain (Kuipers and Moffa, 2017; Niinimäki et al., 2016), or its local moves (Mansinghka et al., 2006; Eaton and Murphy, 2007a; Kuipers et al., 2021). Recently, Viinikka et al. (2020) incorporated many of these advances into an efficient MCMC sampler called *Gadget*.

**Variational Inference**   In the context of structure learning, applying the recent advances in approximate inference based on gradient methods can be difficult due to the discrete nature of the problem (Lorch et al., 2021). Cundy et al. (2021) decomposed the adjacency matrix of a DAG into a triangular matrix and a permutation, and used a continuous relaxation to parametrize a distribution over permutations. Other methods (Annadani et al., 2021; Lorch et al., 2021) encode the acyclicity constraint into a soft prior $P(G)$, based on continuous characterizations of acyclicity (Zheng et al., 2018). While the effect of this prior can be made arbitrarily strong, this does not guarantee that the graphs sampled from the resulting distribution are acyclic. By contrast, our approach guarantees by construction that the support of the posterior approximation is exactly the space of DAGs.

**Sequential decisions**   In this work, we treat the construction of a sample graph from the posterior as a sequential decision problem, starting from the empty graph and adding one edge at a time. Li et al. (2018) use a similar process for creating a generative model over graphs with a fixed ordering over nodes. Similarly, although they do not consider a distribution over graphs, Buesing et al. (2020) use a variant of Monte Carlo Tree Search to approximate a distribution over a pre-specified ordering of discrete random variables. Our method, based on Generative Flow Networks, does not make any assumption on the order in which the edges are added, and multiple edge insertion sequences may lead to the same DAG. Zhu et al. (2020) learn a single high-scoring structure using RL; however, unlike our approach, the creation of this graph does not involve sequential decisions.

## 3   BACKGROUND

A *Bayesian network* is a probabilistic model over $d$ random variables $\{X_1, \ldots, X_d\}$, whose joint distribution factorizes according to a DAG $G$ as

$$P(X_1, \ldots, X_d) = \prod_{k=1}^{d} P\big(X_k \mid \mathrm{Pa}_G(X_k)\big), \qquad (1)$$

where $\mathrm{Pa}_G(X)$ is the set of parents of node $X$ in $G$. Similarly, we denote by $\mathrm{Ch}_G(X)$ the children of $X$; when the context is clear, we may drop the explicit dependency on $G$.

### 3.1   GENERATIVE FLOW NETWORKS

Originally introduced to encourage the discovery of diverse modes of an unnormalized distribution (Bengio et al., 2021a), *Generative Flow Networks* (GFlowNets; Bengio et al., 2021b) are a class of generative models over a discrete and structured sample space $\mathcal{X}$. The structure of a GFlowNet is defined by a DAG over some states $s \in \mathcal{S}$; in general, the sample space over which we wish to define a distribution is only a subset of the overall state space of the GFlowNet: $\mathcal{X} \subseteq \mathcal{S}$. Samples $s \in \mathcal{X}$ are constructed sequentially by following the edges of the DAG, starting from a fixed initial state $s_0$. We also define a special absorbing state $s_f$, called the terminal state, indicating when the sequential construction terminates; some of the states $s \in \mathcal{X}$ are connected to $s_f$, and we call them *complete states*.[2] For example, Bengio et al. (2021a) use a GFlowNet to define a distribution over molecules, where $\mathcal{X}$ would correspond to the space of all (complete) molecules, which are constructed piece by piece by attaching a new block to an atom in a possibly partially constructed molecule (i.e. a state in $\mathcal{S}\backslash\mathcal{X}$).

---

[1] https://github.com/tristandeleu/jax-dag-gflownet

[2]"Complete" here means that the state is a valid sample from the distribution induced by the GFlowNet. This must not be confused with a "complete graph", where all the nodes are connected to one another, when the states are DAGs (see Section 4).

Another example of a GFlowNet structure is given in Fig. 1, illustrating the sequential process of constructing a DAG. A GFlowNet is structurally equivalent to a Markov Decision Process (MDP; Puterman, 1994) with deterministic dynamics, or a Markov Reward Process (Howard, 1971).

In addition to the DAG structure over states, every complete state $s \in \mathcal{X}$ is associated with a *reward $R(s) \geq 0$*, indicating a notion of "preference" for certain states. By convention, $R(s) = 0$ for any incomplete state $s \in \mathcal{S} \backslash \mathcal{X}$. The goal of a GFlowNet is to find a *flow* that satisfies, for all states $s' \in \mathcal{S}$, the following *flow-matching condition*:

$$\sum_{s \in \mathrm{Pa}(s')} F_\theta(s \to s') - \sum_{s'' \in \mathrm{Ch}(s')} F_\theta(s' \to s'') = R(s'), \quad (2)$$

where $F_\theta(s \to s') \geq 0$ is a scalar representing the flow from state $s$ to $s'$, typically parametrized by a neural network. Putting it in words, the overall flow going into $s'$ is equal to the flow going out of $s'$, plus some residual $R(s')$. To learn the parameters $\theta$ of the flow with SGD, we can turn (2) into a regression problem, e.g. using a least squares objective over sampled states.

If the conditions in (2) are satisfied for all states $s'$, a GFlowNet induces a generative process to sample complete states $s \in \mathcal{X}$ with probability proportional to $R(s)$. Starting from the initial state $s_0$, if we sample a complete trajectory $(s_0, s_1, \ldots, s_T, s, s_f)$ using the transition probability defined as the normalized outgoing flow

$$P(s_{t+1} \mid s_t) \propto F_\theta(s_t \to s_{t+1}), \quad (3)$$

with the conventions $s_{T+1} = s$ and $s_{T+2} = s_f$, then $s$ is sampled with probability $P(s) \propto R(s)$. Note that the linear system in (2) is in general underdetermined, and therefore it may admit many solutions $F_\theta(s \to s')$ that all induce the same distribution $\propto R(s)$. Unlike MCMC, each sample $s \in \mathcal{X}$ is constructed from scratch, starting at the initial state $s_0$, instead of traversing $\mathcal{X}$ from sample to sample. Therefore, the underlying Markov process of the GFlowNet does not have to be irreducible, which is typically necessary in MCMC, but merely requires all the complete states to be reachable from the initial state. Finally, although GFlowNets borrow terminology from RL and control theory (e.g. MDPs, rewards, trajectories), their objective is different from the typical RL training objective: the latter seeks to maximize a function of the rewards, while the goal of GFlowNets is to model the whole distribution proportional to the rewards.

## 3.2 DETAILED-BALANCE CONDITION

Since the flows are added together, one of the downsides of the flow-matching condition is that flows tend to be orders of magnitude larger the closer we are of the initial state (Bengio et al., 2021a), making it challenging to parametrize $F_\theta$. Bengio et al. (2021b) proposed an alternative characterization of GFlowNets inspired by the detailed-balance

equations from the literature on Markov chains (Grimmett and Stirzaker, 2020). Instead of working with flows, this condition uses a parametrization of the forward transition probability $P_\theta(s_{t+1} \mid s_t)$ directly, together with a backward transition probability $P_B(s_t \mid s_{t+1})$ to enforce reversibility. As opposed to $P_\theta(s_{t+1} \mid s_t)$, which is a distribution over the children of $s_t$, $P_B(s_t \mid s_{t+1})$ is a distribution over the parents of $s_{t+1}$ in the structure of the GFlowNet. If all the states of the GFlowNet are complete (except the terminal state $s_f$), which will be the case here for generating DAGs, then we show in Appendix B that we can write the *detailed-balance condition* for all transitions $s \to s'$ as follows:

$$R(s') P_B(s \mid s') P_\theta(s_f \mid s) = R(s) P_\theta(s' \mid s) P_\theta(s_f \mid s').$$

Similar to Section 3.1, finding $P_\theta$ and $P_B$ that satisfy this condition for all the transitions $s \to s'$ of the GFlowNet also yields a sampling process of complete states $s$ with probability proportional to $R(s)$, based on the forward transition probability $P_\theta(s_{t+1} \mid s_t)$. Because this system of equations also admits many solutions, similar to (2), we can set the backward transition probability $P_B$ to some fixed distribution (e.g. the uniform distribution over the parent states) to reduce the search space, making $P_\theta$ the only quantity to learn and, with enough capacity (to satisfy the constraints), there is a unique solution $P_\theta$ (Bengio et al., 2021b).

To fit the parameters $\theta$ of the forward transition probability, we can minimize the following non-linear least squares objective for all the transitions $s \to s'$ of the GFlowNet, called the *detailed-balance loss*:

$$\mathcal{L}(\theta) = \sum_{s \to s'} \left[ \log \frac{R(s') P_B(s \mid s') P_\theta(s_f \mid s)}{R(s) P_\theta(s' \mid s) P_\theta(s_f \mid s')} \right]^2. \quad (4)$$

Alternatively, we can minimize this loss in expectation, using a distribution $\pi(s \to s')$ with full support over transitions (i.e. for all transitions $s \to s'$ in the GFlowNet, we have $\pi(s \to s') > 0$; see Section 5.2).

## 4 GFLOWNET OVER DIRECTED ACYCLIC GRAPHS

Our objective in this paper is to construct a distribution over DAGs. This is a challenging problem in general, as the space of DAGs is discrete and combinatorially large. We propose to use a GFlowNet to model such a distribution; this is particularly appropriate here since graphs are composite objects, and the acyclicity constraint can be obtained by constraining the allowed actions in each state (as in Figure 1). Note that the DAGs in this section and thereafter represent the states of the GFlowNet, and they must not be confused with the DAG structure of the GFlowNet itself.

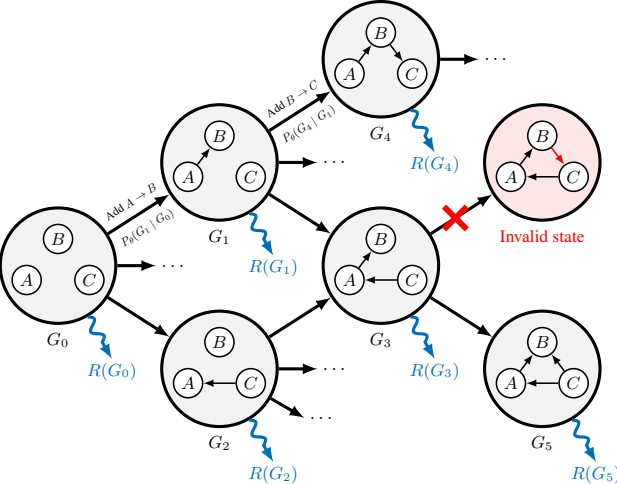

Figure 1: Structure of a GFlowNet over DAGs. The states of the GFlowNet correspond to DAGs, with the initial state $G_0$ being the completely disconnected graph. Each state $G$ is complete (i.e. connected to the terminal state $s_f$, represented by blue arrows for brevity) and associated to a reward $R(G)$. Transitioning from one state to another corresponds to adding an edge to the graph. The state in red is invalid since the graph includes a cycle.

## 4.1 STRUCTURE OF THE GFLOWNET

We consider a GFlowNet where the states are DAGs over $d$ (labeled) nodes. Since the states of the GFlowNet are graphs, we will use the notation $G$ to denote a state, in favour of $s$ as in Section 3.1, except for the terminal state $s_f$. A transition $G \rightarrow G'$ in this GFlowNet corresponds to adding an edge to $G$ to obtain the graph $G'$; in other words, the graphs are constructed one edge at a time, starting from the initial state $G_0$, which is the fully disconnected graph over $d$ nodes. Since we assume that all the states $G$ of the GFlowNet are valid DAGs, they are all complete (i.e. connected to the terminal state $s_f$) with a corresponding reward $R(G)$. Figure 1 shows an illustration of the structure of such a GFlowNet, where the states are DAGs over $d = 3$ nodes. This application to graphs highlights the importance of the DAG structure of the GFlowNet, since there can be multiple paths leading to the same state: for any graph $G$ with $k$ edges, there are $k!$ possible paths from $G_0$ leading to $G$, because the edges of $G$ may have been added in any order.

To guarantee the integrity of the GFlowNet, we have to ensure that adding a new edge to some state $G$ also yields a valid DAG, meaning that this edge (1) must not be already present in $G$, and (2) must not introduce a cycle. Fortunately, we can filter out invalid actions using some mask $m$ associated to the graph, built from the adjacency matrix of $G$ and the transitive closure of its transpose, and that can be updated efficiently after the addition an edge (Giudici and Castelo, 2003). A description of this update is given in Appendix C for completeness.

## 4.2 FORWARD TRANSITION PROBABILITIES

Following Section 3.2, the GFlowNet may be parametrized only by the forward transition probabilities $P_\theta(G_{t+1} \mid G_t)$; here, $G_{t+1}$ might be the terminal state $s_f$ by abuse of notation. To make sure that the detailed-balance conditions can be satisfied, we need to define these transition probabilities using a sufficiently expressive function, such as a neural network. We use a hierarchical model, where the forward transition probabilities are defined using two neural networks: (1) a network modeling the probability of terminating $P_\theta(s_f \mid G)$, and (2) another giving the probability $P_\theta(G' \mid G, \neg s_f)$ of transitioning to a new graph $G'$, given that we do not terminate. The probability of taking a transition $G \rightarrow G'$ is then given by

$$P_\theta(G' \mid G) = \big(1 - P_\theta(s_f \mid G)\big)P_\theta(G' \mid G, \neg s_f). \quad (5)$$

In practice, as $G'$ is the result of adding an edge to the DAG $G$, we can model $P_\theta(G' \mid G, \neg s_f)$ as a probability distribution over the $d^2$ possible edges one could add to $G$—this includes self-loops, for simplicity, even though these actions are guaranteed to be invalid. We can use the mask $m$ introduced in Section 4.1 to filter out actions that would not lead to a valid DAG $G'$ and set $P_\theta(G' \mid G, \neg s_f) = 0$ for any invalid action (as well as normalize $P_\theta$ accordingly).

## 4.3 PARAMETRIZATION WITH LINEAR TRANSFORMERS

Beyond having enough capacity to satisfy as well as possible the detailed-balance condition at all states, we choose to parametrize the forward transition probabilities with neural networks to benefit from their capacity to generalize to states not encountered during training. In practice, instead of defining two separate networks to parametrize $P_\theta(s_f \mid G)$ and $P_\theta(G' \mid G, \neg s_f)$, we use a single neural network with a common backbone and two separate heads, to benefit from parameter sharing. The full architecture is given in Figure 2.

Our choice of neural network architecture is motivated by multiple factors: we want an architecture (1) that is invariant to the order of the inputs, since $G$ is represented as a set of edges, (2) that transforms a set of input edges into a set of output probabilities for each edge to be added, in order to define $P_\theta(G' \mid G, \neg s_f)$, and (3) whose parameters $\theta$ do not scale too much with $d$. A natural option would be to use a Transformer (Vaswani et al., 2017); however, because the size of our inputs is $d^2$, the self-attention layers would scale as $d^4$, and this would severely limit our ability to apply our method to model a distribution over larger DAGs.

We opted for a Linear Transformer (Katharopoulos et al., 2020) instead, which has the advantage to not suffer from this quadratic scaling in the input size. This architecture

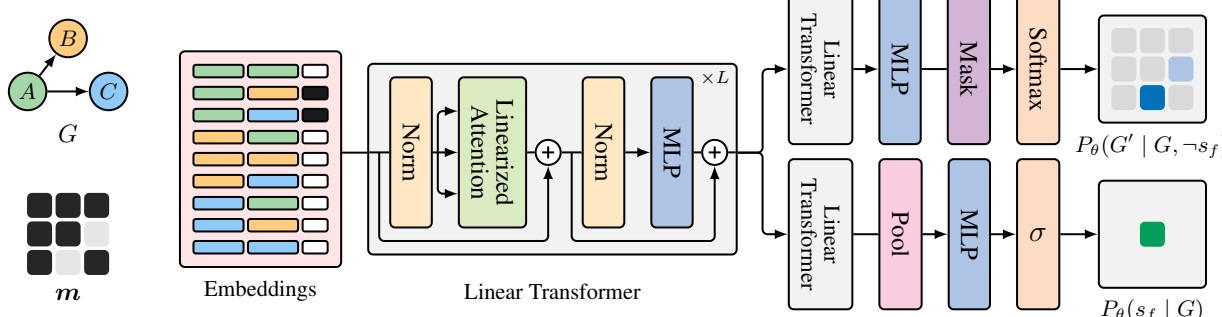

Figure 2: Neural network architecture of the forward transition probabilities $P_\theta(G_{t+1} \mid G_t)$. The input graph $G$ is encoded as a set of $d^2$ possible edges (including self-loops). Each directed edge is embedded using the embeddings of its source and target, with an additional vector indicating whether the edge is present in $G$. These embeddings are fed into a Linear Transformer (Katharopoulos et al., 2020), with two separate output heads. The first head (above) gives the probability to add a new edge $P_\theta(G' \mid G, \neg s_f)$, using the mask $m$ associated to $G$ to filter out invalid actions; here, the only valid actions are either adding $B \to C$, or $C \to B$. The second head (below) gives the probability to terminate the trajectory $P_\theta(s_f \mid G)$.

relies on a linearized attention mechanism, defined as

$$Q = \boldsymbol{x}W_Q \qquad K = \boldsymbol{x}W_K \qquad V = \boldsymbol{x}W_V$$

$$\text{LinAttn}_k(\boldsymbol{x}) = \frac{\sum_{j=1}^{J} \big(\phi(Q_k)^\top \phi(K_j)\big) V_j}{\sum_{j=1}^{J} \phi(Q_k)^\top \phi(K_j)}, \qquad (6)$$

where $\boldsymbol{x}$ is the input of the linearized attention layer, $\phi(\cdot)$ is a non-linear feature map, $J$ is the size of the input $\boldsymbol{x}$ (in our case, $J = d^2$), and $Q$, $K$, and $V$ are linear transformations of $\boldsymbol{x}$ corresponding to the queries, keys, and values respectively, as is standard with Transformers.

# 5 APPLICATION TO BAYESIAN STRUCTURE LEARNING

We are given a dataset $\mathcal{D} = \{\boldsymbol{x}^{(1)}, \dots, \boldsymbol{x}^{(N)}\}$ of $N$ observations $\boldsymbol{x}^{(j)}$, each consisting of $d$ elements. We consider the task of characterizing the posterior distribution $P(G \mid \mathcal{D})$ over Bayesian networks that model these observations. We assume that the samples in $\mathcal{D}$ are iid. and fully-observed. As an alternative to MCMC (Madigan et al., 1995) or variational inference (Lorch et al., 2021), we approximate the posterior distribution over DAGs using a GFlowNet, as described in the previous section. For any DAG $G$, we will define its reward as the joint probability

$$R(G) = P(G)P(\mathcal{D} \mid G), \qquad (7)$$

where $P(G)$ is a prior over DAGs (Eggeling et al., 2019), and $P(\mathcal{D} \mid G)$ is the marginal likelihood. In Sec. 3.2, we saw that if the detailed-balance conditions are satisfied for all the states of the GFlowNet, then this yields a sampling process with probability proportional to $R(G)$. Therefore, by Bayes' theorem, a GFlowNet with the specific reward function in (7) approximates the posterior distribution $P(G \mid \mathcal{D}) \propto R(G)$. We call our method *DAG-GFlowNet*.

## 5.1 MODULARITY & COMPUTATIONAL EFFICIENCY

Following prior works on Bayesian structure learning, we assume that both the priors over parameters $P(\phi \mid G)$ of the Bayesian network (required to compute the marginal likelihood) and over structures $P(G)$ are *modular* (Heckerman et al., 1995; Chickering et al., 1995). As a consequence the reward $R(G)$ is also modular, and its logarithm can be written as a sum of local scores that only depend on individual variables and their parents in $G$:

$$\log R(G) = \sum_{j=1}^{d} \text{LocalScore}\big(X_j \mid \text{Pa}_G(X_j)\big). \qquad (8)$$

Note that with our choice of reward, $\log R(G)$ corresponds to the Bayesian score (Koller and Friedman, 2009). Examples of modular scores include the BDe score (Heckerman et al., 1995) and the BGe score (Geiger and Heckerman, 1994; Kuipers et al., 2014). In order to fit the parameters $\theta$ of the GFlowNet, we will use the detailed-balance loss in (4). We can observe that this loss function only involves the difference in log-rewards $\log R(G') - \log R(G)$ between two consecutive states, where $G'$ is the result of adding some edge $X_i \to X_j$ to the DAG $G$. Using our assumption of modularity, we can therefore compute this difference efficiently, as the terms in (8) remain unchanged for $j' \neq j$:

$$\log R(G') - \log R(G) = \text{LocalScore}\big(X_j \mid \text{Pa}_G(X_j) \cup \{X_i\}\big)$$
$$- \text{LocalScore}\big(X_j \mid \text{Pa}_G(X_j)\big). \qquad (9)$$

This difference in local scores is sometimes called the *delta score*, or the *incremental value* (Friedman and Koller, 2003), and has been employed in the literature to improve the efficiency of search algorithms (Chickering, 2002; Koller and Friedman, 2009).

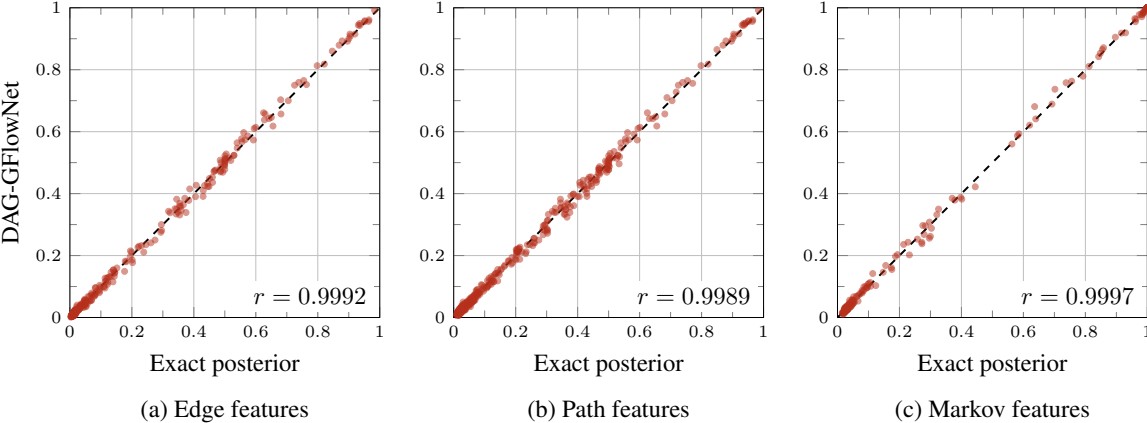

|  (a) Edge features | (b) Path features | (c) Markov features |

Figure 3: Comparison between the exact posterior distribution and the posterior approximation from DAG-GFlowNet, for different structural features: (a) edge features $X_i \rightarrow X_j$, (b) path features $X_i \rightsquigarrow X_j$, (c) Markov features $X_i \sim_M X_j$. Each point corresponds to a feature computed for specific variables $X_i$ and $X_j$ in a graph over $d = 5$ nodes, either based on the exact posterior (x-axis), or the posterior approximation found with the GFlowNet (y-axis). We repeated this experiment with 20 different (ground-truth) DAGs. The Pearson correlation coefficient $r$ is included in the bottom-right corner of each plot.

## 5.2 OFF-POLICY LEARNING

As the number of states in DAG-GFlowNet is super-exponential in $d$, the number of nodes in each DAG $G$, it would be impractical to minimize the detailed-balance loss for all possible transitions $G \rightarrow G'$. Alternatively, we can minimize this loss in expectation using a distribution $\pi(G \rightarrow G')$ with full support over transitions:

$$\mathcal{L}(\theta) = \mathbb{E}_\pi \left[ \left[ \log \frac{R(G')P_B(G \mid G')P_\theta(s_f \mid G)}{R(G)P_\theta(G' \mid G)P_\theta(s_f \mid G')} \right]^2 \right]. \tag{10}$$

This distribution $\pi(G \rightarrow G')$ can be arbitrary; for example, we can use $P_\theta(G' \mid G)$ directly and learn it *on-policy* (Rummery and Niranjan, 1994), as long as it assigns non-zero probability to any next state $G'$.

Taking inspiration from Deep Q-learning (Mnih et al., 2015), we instead learn $P_\theta$ using *off-policy* data. Transitions $G \rightarrow G'$ are collected based on $P_\theta(G' \mid G)$, along with their corresponding delta score ((9)), and they are stored in a replay buffer. We can also sample some transitions uniformly at random, with probability $\varepsilon$, to encourage exploration. To estimate $\mathcal{L}(\theta)$ and update the parameters $\theta$, we can then sample a mini-batch of transitions randomly from the replay buffer. Moreover, again inspired by Deep Q-learning (Van Hasselt et al., 2018), we found it advantageous to evaluate $P_{\bar\theta}(s_f \mid G')$ in (10) with a separate target network—where the parameters $\bar\theta$ are updated periodically.

## 6 EXPERIMENTAL RESULTS

We compared DAG-GFlowNet against 3 broad classes of Bayesian structure learning algorithms: MCMC, non-

parametric DAG Bootstrapping (Friedman et al., 1999), and variational inference. We used Structure MCMC (MC[3]; Madigan et al., 1995) and the recent Gadget (Viinikka et al., 2020) samplers as two representative methods based on MCMC. Following Lorch et al. (2021), we used two variants of Bootstrapping based on the score-based algorithm GES (Bootstrap GES; Chickering, 2002), and the constraint-based algorithm PC (Bootstrap PC; Spirtes et al., 2000), as the internal structure learning routines. Finally for methods based on variational inference, we used DiBS (Lorch et al., 2021) and BCD Nets (Cundy et al., 2021). Throughout this section, we used the BGe score for continuous data, and the BDe score for discrete data, to compute $\log p(\mathcal{D} \mid G)$.

### 6.1 COMPARISON WITH THE EXACT POSTERIOR

In order to measure the quality of the posterior approximation returned by DAG-GFlowNet, we want to compare it with the exact posterior distribution $P(G \mid \mathcal{D})$. However, the latter requires an exhaustive enumeration of all possible DAGs, which is only feasible for graphs with no more than 5 nodes. Therefore, we sampled $N = 100$ datapoints from a randomly generated (under an Erdős-Rényi model; Erdős and Rényi, 1960) linear-Gaussian Bayesian network over $d = 5$ variables. We used the BGe score to compute the reward $R(G) = P(G)P(\mathcal{D} \mid G)$. The exact posterior distribution $P(G \mid \mathcal{D})$ is obtained by enumerating all 29,281 possible DAGs over 5 nodes and computing their respective rewards $R(G)$ (normalized to sum to 1).

We evaluated the quality of the approximation based on the probability of various structural features. For example, using samples $\{G_1, G_2, \ldots, G_n\}$ from the posterior approxima-

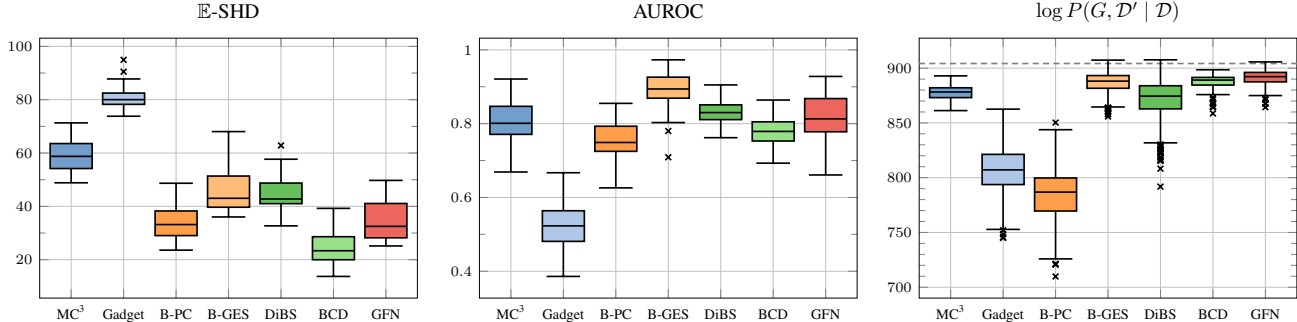

Figure 4: Bayesian structure learning of linear-Gaussian Bayesian networks with $d = 20$ nodes. Results for $\mathbb{E}$-SHD & AUROC are aggregated over 25 randomly generated datasets $\mathcal{D}$, sampled from different (ground-truth) Bayesian networks. Results for $\log P(G, \mathcal{D}' \mid \mathcal{D})$ are given for a single dataset $\mathcal{D}$; the dashed line corresponds to the log-likelihood of the ground truth graph $G^\star$. For $\mathbb{E}$-SHD lower is better, and for AUROC and $\log P(G, \mathcal{D}' \mid \mathcal{D})$ higher is better. Labels: B-PC = Bootstrap-PC, B-GES = Bootstrap-GES, BCD = BCD Nets, GFN = DAG-GFlowNet.

tion, the marginal probability of an *edge feature* $X_i \rightarrow X_j$ can be estimated with

$$P_\theta(X_i \rightarrow X_j \mid \mathcal{D}) \approx \frac{1}{n} \sum_{k=1}^{n} \mathbf{1}(X_i \rightarrow X_j \in G_k), \quad (11)$$

where $\mathbf{1}(\cdot)$ is the indicator function. For the exact posterior, we can obtain the posterior probability of the edge feature by simply marginalizing over $P(G \mid \mathcal{D})$. Similarly, we compute (or estimate) the marginal probability of a *path feature* $X_i \rightsquigarrow X_j$, i.e. of a (directed) path existing from $X_i$ to $X_j$, and the probability of a *Markov feature* $X_i \sim_M X_j$, i.e. of $X_i$ being in the Markov blanket of $X_j$ (Friedman and Koller, 2003). These features are computed for all variables $X_i$ and $X_j$ in the Bayesian network.

In Figure 3, we compare the probabilities of these features for both the exact posterior and the distribution induced by DAG-GFlowNet, where we repeated the experiment above with 20 different (ground-truth) Bayesian networks. We observe that the probabilities of all structural features estimated by the GFlowNet are strongly correlated with the exact marginal probabilities. This shows that DAG-GFlowNet is capable of learning a very accurate approximation of the posterior distribution over graphs $P(G \mid \mathcal{D})$.

### 6.2 SIMULATED DATA

We follow the experimental setup of Zheng et al. (2018) & Lorch et al. (2021), and sample synthetic data from linear-Gaussian Bayesian networks with randomly generated structures; details about this data generation process are given in Appendix D.2. To show that DAG-GFlowNet can effectively approximate the posterior distribution over larger graphs, we experimented with Bayesian networks of size $d = 20$ (and $d = 50$, see Appendix D.2). Similar to Section 6.1, the ground-truth graphs are sampled according to an Erdős-Rényi model, with $2d$ edges in expectation—a

setting sometimes referred to as ER2 (Cundy et al., 2021). For each experiment, we sampled a dataset $\mathcal{D}$ of $N = 100$ observations, and we used the BGe score to compute $R(G)$.

Since we have access to the ground-truth graph $G^\star$ that generated $\mathcal{D}$, we evaluate the performance of each algorithm with the *expected structural Hamming distance* ($\mathbb{E}$-SHD) to $G^\star$ over the posterior approximation; a detailed definition is available in Appendix D.1. We also compute the *area under the ROC curve* (AUROC; Husmeier, 2003) for the edge features as defined in (11), compared to the edges of $G^\star$. Finally, we compute the joint log-likelihood $\log P(G, \mathcal{D}' \mid \mathcal{D})$ on a held-out dataset $\mathcal{D}'$; we chose this metric over the log-predictive likelihood $\log P(\mathcal{D}' \mid \mathcal{D})$, as proposed by Eaton and Murphy (2007a), to study the effect of the posterior approximation $P(G \mid \mathcal{D})$.

The results on graphs with $d = 20$ nodes are shown in Figure 4. We observe that both in terms of $\mathbb{E}$-SHD & AUROC, DAG-GFlowNet, is competitive against all other methods, in particular those based on MCMC, and this does not come at a cost in terms of its predictive capacity on held-out data. In particular, we can see that the distribution induced by DAG-GFlowNet yields a predictive log-likelihood concentrated near the log-likelihood of the ground-truth DAG $G^\star$.

### 6.3 APPLICATION: FLOW CYTOMETRY DATA

We also evaluated DAG-GFlowNet on real-world flow cytometry data (Sachs et al., 2005) to learn protein signaling pathways. The data consists of continuous measurements of $d = 11$ phosphoproteins in individual T-cells. Out of all the measurements, we selected the $N = 853$ observations corresponding to the first experimental condition of Sachs et al. (2005) as our dataset $\mathcal{D}$. Following prior work on structure learning, we used the DAG inferred by Sachs et al. (2005), containing $d = 11$ nodes and 17 edges, as our graph of reference (ground-truth). However, it should be noted

that this "consensus graph" may not represent a realistic and complete description of the system being modeled here (Mooij et al., 2020). We standardized the data, and used the BGe score to compute $R(G)$.

Table 1: Learning protein signaling pathways from flow cytometry data (Sachs et al., 2005). All results include a 95% confidence interval estimated with bootstrap resampling.

|  | $\mathbb{E}$-# Edges | $\mathbb{E}$-SHD | AUROC |
|---|---|---|---|
| MC[3] | $10.96 \pm 0.09$ | $22.66 \pm 0.11$ | 0.508 |
| Gadget | $10.59 \pm 0.09$ | $21.77 \pm 0.10$ | 0.479 |
| Bootstrap GES | $11.11 \pm 0.09$ | $23.07 \pm 0.11$ | **0.548** |
| Bootstrap PC | $7.83 \pm 0.04$ | $20.65 \pm 0.06$ | 0.520 |
| DiBS | $12.62 \pm 0.16$ | $23.32 \pm 0.14$ | 0.518 |
| BCD Nets | $4.14 \pm 0.09$ | **18.14 ± 0.09** | 0.510 |
| DAG-GFlowNet | $11.25 \pm 0.09$ | $22.88 \pm 0.10$ | 0.541 |

In Table 1, we compare the expected SHD and the AUROC obtained with DAG-GFlowNet and other approaches. While BCD Nets and Bootstrap PC have a smaller $\mathbb{E}$-SHD, suggesting that the distribution is concentrated closer to the consensus graph, in reality they tend to be more conservative and sample graphs with fewer edges. Overall, DAG-GFlowNet offers a good trade-off between performance (as measured by the $\mathbb{E}$-SHD and the AUROC), and getting a distribution that assigns higher probability to DAGs with more edges. We also observed that $1.50\%$ of the graphs sampled with DiBS contained a cycle.

Beyond these metrics, we would like to test if the advantages of Bayesian structure learning are also reflected in the distribution induced by DAG-GFlowNet. In particular, we want to study (1) if this distribution covers multiple high-scoring DAGs, instead of being peaked at a single most likely graph, and (2) if the GFlowNet can sample a variety of DAGs from the same Markov equivalence class (MEC), showing the inherent uncertainty over equivalent graphs. In Figure 5, we visualize the MECs of the graphs sampled with DAG-GFlowNet, and two methods based on MCMC (MC[3] and Gadget); other baselines were excluded for clarity. The size of each point represents the number of unique DAGs in the corresponding MEC. We observe that DAG-GFlowNet largely follows the behavior of MCMC: the distribution does not collapse to a single most-likely DAG, and covers multiple MECs. Moreover, the GFlowNet is also capable of sampling different equivalent DAGs (corresponding to larger points), showing again that the distribution does not collapse to a single representative of the MECs with higher marginal probability. We also observe that the maximum a posteriori MEC found by DAG-GFlowNet reaches a higher score than the one found with Gadget, but a lower score than MC[3]; as a point of reference, the score of the best MEC obtained with GES (Chickering, 2002) is $-10,716.12$.

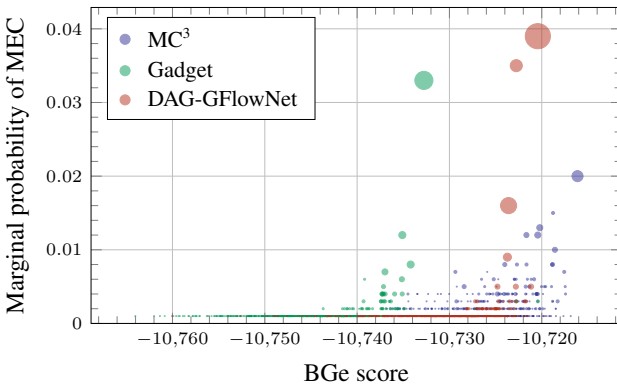

Figure 5: Coverage of the posterior approximations learned on flow cytometry data (Sachs et al., 2005). Each point corresponds to a sampled Markov equivalence class, and its size represents the number of different DAGs (in the equivalence class) sampled from the posterior approximation. See Figure 8 in Appendix D.3 for an additional comparison with methods based on Variational Inference.

## 6.4 APPLICATION: INTERVENTIONAL DATA

In addition to the observational data we used in Section 6.3, Sachs et al. (2005) also provided flow cytometry data under different experimental conditions, where the T-cells were perturbed with some reagents; this effectively corresponds to interventional data (Pearl, 2009). Although a molecular intervention may be imperfect and affect multiple proteins (Eaton and Murphy, 2007b), we assume here that these interventions are perfect, and the intervention targets are known. We used a discretized dataset of $N = 5,400$ samples from 9 experimental conditions—of which 6 are interventions. We modified the BDe score to handle this mixture of observational and interventional data (Cooper and Yoo, 1999).

Table 2: Combining discrete interventional and observational flow cytometry data (Sachs et al., 2005). ⋆Result reported in Eaton and Murphy (2007b).

|  | $\mathbb{E}$-# Edges | $\mathbb{E}$-SHD | AUROC |
|---|---|---|---|
| Exact posterior⋆ | — | — | **0.816** |
| MC[3] | $25.97 \pm 0.01$ | **25.08 ± 0.02** | 0.665 |
| DAG-GFlowNet | $30.66 \pm 0.04$ | $27.77 \pm 0.03$ | 0.700 |

In Table 2, we compare with Eaton and Murphy (2007b), which compute the AUROC of the exact posterior using dynamic programming, therefore working as an upper bound for what a posterior approximation can achieve. They achieve this at the expense of computing only edge marginals, without providing access to a distribution over DAGs. We also use the modified BDe score with MC[3], which predicts sparser graphs with higher SHD than DAG-

GFlowNet, but lower AUROC. Note that this setup is different from previous works which use continuous data instead (Brouillard et al., 2020; Faria et al., 2022).

# 7 CONCLUSION

We have proposed a new method for Bayesian structure learning, based on a novel class of probabilistic models called GFlowNets, where the generation of a sample graph is treated as a sequential decision problem. We introduced a number of enhancements to the standard framework of GFlowNets, specifically designed for approximating a distribution over DAGs. In cases where the data is limited and measuring the epistemic uncertainty is critical, DAG-GFlowNet offers an effective solution to approximate the posterior distribution over DAGs $P(G \mid \mathcal{D})$. However, we also observed that in its current state, DAG-GFlowNet may suffer from some limitations, notably as the size of the dataset $\mathcal{D}$ increases; see Appendix A for a discussion.

While DAG-GFlownet operates on the space of DAGs directly, the structure of the GFlowNet may eventually be adapted to work with alternative representations of statistical dependencies in Bayesian networks, such as essential graphs for MECs (Chickering, 2002). Moreover, although we have already shown that DAG-GFlowNet can approximate the posterior using a mixture of observational and interventional data, we will continue to study in future work its applications to causal discovery, especially in the context of learning the structure of models with latent variables.

### Acknowledgements

We would like to thank Emmanuel Bengio, Paul Bertin, and Valentin Thomas for the useful discussions about the project, and Dinghuai Zhang, Kolya Malkin, and Xu Ji for their valuable feedback on the paper. This research was partially supported by the Canada CIFAR AI Chair Program and by Samsung Electronics Co., Ldt. Simon Lacoste-Julien is a CIFAR Associate Fellow in the Learning in Machines & Brains program, Yoshua Bengio is a CIFAR Senior Fellow and Stefan Bauer is a CIFAR Azrieli Global Scholar.

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
