# OpenReview forum: "Bayesian Structure Learning with Generative Flow Networks"
_auai.org/UAI/2022/Conference — UAI 2022 Poster_

### Official Review · Reviewer_j5gU · 2022-04-12

**Q2(1) Originality/Novelty:** 2
**Q2(2) Significance/Impact:** 2
**Q2(3) Correctness/Technical Quality:** 3
**Q2(6) Clarity Of Writing:** 3
**Q6 Overall Score:** 4
**Q8 Confidence In Your Score:** 3

**Q1 Summary And Contributions:**

This paper proposes a probabilistic model called DAG-GFlowNet to approximate the posterior distribution over the structure (DAGs) of Bayesian network, which improves the original generative flow networks with a flow-matching condition and corresponding loss function, a hierarchical probabilistic model for forward transitions and additional reinforcement learning tools. Experiments conducted on simulated and real data show the effectiveness of the proposed method.

**Q2 Assessment Of The Paper:**

More detailed information regarding each of these aspects is given below:

**Q2(4) Quality Of Experiments (Optional):**

2: Fair: The experimental evaluation is weak: important baselines are missing, or the results do not adequately support the main claims.

**Q2(5) Reproducibility:**

2: Fair: Key resources (e.g., proofs, code, data) are unavailable but key details (e.g., proof sketches, experimental setup) are sufficiently well-described for an expert to confidently reproduce the main results.

**Q3 Main Strengths:**

1. The paper presents a GFlowNet as an alternative to MCMC for approximating the posterior distribution over the structure of Bayesian networks.

2. The paper presents a sum of local sores that only depends on individual variables and their parents in G by combing the reward R(G) with Bayesian score.


**Q4 Main Weakness:**

1. The contributions of the paper are not explained clearly in the introduction.

2. The process of Bayesian structure learning usually includes adding, removing and reversing edges in DAG, but GFlowNets only performs adding operations of edges. This might make the method inefficient since the GFlowNet must be large enough to contain all possible DAGs. Moreover, the time complexity analysis of the proposed method is not given.

3. In the experiments, classical Bayesian structure learning methods were not chosen for comparison and no efficiency tests. Furthermore, the experimental results are not convinced and the improvements are not significant.


**Q5 Detailed Comments To The Authors:**

One of my biggest concerns is the experiment:
(1) Except MCMC, non-parametric DAG Bootstrapping and variational inference, some classical Bayesian structure learning algorithms, such as K2 algorithm, hill climbing algorithm, should be adopted for comparison.
(2) The paper has discussed the computational efficiency of Bayesian structure learning, but the experiments only test the effectiveness of the proposed method.
(3) In Figure 4 and Table 1, the E-SHD and AUROC of DAG-GFlowNet are worse than other methods. This should be discussed in detail.
(4) The datasets should be described more clearly. For example, how to obtain the 20 different (ground-truth) Bayesian networks in Section 6.1? How to sample synthetic data from linear-Gaussian Bayesian networks in Section 6.2


**Q7 Justification For Your Score:**

Although using generative flow networks for Bayesian network learning is interesting, the experimental results are not convinced and should be improved. Thus, my recommendation is borderline rejection.

**Q9 Complying With Reviewing Instructions:**

1: Yes.

---

### Official Review · Reviewer_VhH3 · 2022-04-13

**Q2(1) Originality/Novelty:** 3
**Q2(2) Significance/Impact:** 3
**Q2(3) Correctness/Technical Quality:** 3
**Q2(6) Clarity Of Writing:** 3
**Q6 Overall Score:** 7
**Q8 Confidence In Your Score:** 4

**Q1 Summary And Contributions:**

The paper proposes to adopt the generative flow networks to learn the posterior distribution over the DAGs representing Bayesian networks.


**Q2 Assessment Of The Paper:**

More detailed information regarding each of these aspects is given below:

**Q2(4) Quality Of Experiments (Optional):**

3: Good: The experimental evaluation is adequate, and the results convincingly support the main claims.

**Q2(5) Reproducibility:**

3: Good: Key resources (e.g., proofs, code, data) are available and key details (e.g., proofs, experimental setup) are sufficiently well-described for competent researchers to confidently reproduce the main results.

**Q3 Main Strengths:**

Studying the challenging problem of learning a distribution over DAGs.


**Q4 Main Weakness:**

Learning complexity of the proposed approach

**Q5 Detailed Comments To The Authors:**

The proposed approach has been evaluated on various problems with simulated and real data, on both discrete and linear-Gaussian Bayesian networks.

The paper is well written and all the notions and concepts are well described.

A discussion on the learning complexity should be reported. It seems that the proposed approach could have some limitations as the size of the dataset increases. Appendix A should be extended providing some solutions.



**Q7 Justification For Your Score:**

The obtained results are good but the complexity of the method should be discussed

**Q9 Complying With Reviewing Instructions:**

1: Yes.

---

### Official Review · Reviewer_X8kv · 2022-04-13

**Q2(1) Originality/Novelty:** 3
**Q2(2) Significance/Impact:** 4
**Q2(3) Correctness/Technical Quality:** 3
**Q2(6) Clarity Of Writing:** 4
**Q6 Overall Score:** 7
**Q8 Confidence In Your Score:** 3

**Q1 Summary And Contributions:**

Summary: use of GFlowNetsfor structural learning in BNs, being GFlowNets from RL and deep generative models. They’re used for modeling a distribution over possible graphs. The construction/search is done sequentially by adding edges.  Contributions: extension & improvement of GFlowNets to better fit the particular problem: flow-matching condition and corresponding loss function, hierarchical probabilistic model for forward transitions, and the use of other tools from Reinforcement Learning.


**Q2 Assessment Of The Paper:**

More detailed information regarding each of these aspects is given below:

**Q2(4) Quality Of Experiments (Optional):**

3: Good: The experimental evaluation is adequate, and the results convincingly support the main claims.

**Q2(5) Reproducibility:**

2: Fair: Key resources (e.g., proofs, code, data) are unavailable but key details (e.g., proof sketches, experimental setup) are sufficiently well-described for an expert to confidently reproduce the main results.

**Q3 Main Strengths:**

The novelty of applying GFlowNets to this particular problem.
The way in which the difficulties have been solved, so that innovative solutions are given.
The paper is well-written and the comparison and plots are relevant.
The experimentation includes artificially created problems and a couple of applications

**Q4 Main Weakness:**

The method seem to slow compared with classical approaches, mostly with 50-nodes dimensionality.
Lack of reproducibility, no code available (if I am not wrong)


**Q5 Detailed Comments To The Authors:**

This work proposes the use of Generative Flow Networks (GFlowNets) for structural learning in Bayesian Networks. GFlowNets are related to reinforcement learning and deep generative models. But they work also with probabilities. Their features are used for modeling a distribution over possible graphs, in order to find the most appropriate for the particular learning problem. The construction/search is done sequentially, starting from the empty graph, by adding one edge at a time.

As contributions:

The authors extend and improve the original GFlowNet to better fit the particular problem to solve: flow-matching condition and corresponding loss function, hierarchical probabilistic model for forward transitions, and the use of other tools from Reinforcement Learning).

The procedure of constructing a distribution over the Directed Acyclic Graphs (DAGs) is a key point.

The main contribution would be the methodology for applying GFlowNets into actual BN learning, modular computation of the reward and off-policy learning for getting the probability on the graphs’ transitions includes both originality but re-using of concepts from other paradigms distinct from the classical PGMs. And, most importantly, they seem to work.

Experimental setup seems correct, the initial five nodes result too small, but it is true that it is extended for 20 and 50. Even though, in some domains these can be still low-dimensional, I think the approach is correctly tested.

The performance is competitive enough, and I really find it a promising research line. Even though the results are not the best ones, it is clear that the measurements are on the level of other well-known techniques.

Minor comments:

Apart from complexity analysis, I miss some study on processing time, comparing with the other methods, as it seems it will be much slower.

I’m not familiarised with AUROC naming, I have always encountered it as auc.


**Q7 Justification For Your Score:**

I did not thoroughly check all the details, but I think the paper is technically sound. I also find it innovative and original, and well developed in the main aspects: presentation of the most important points and good experimentation.


**Q9 Complying With Reviewing Instructions:**

1: Yes.

---

### Official Review · Reviewer_RGiV · 2022-04-18

**Q2(1) Originality/Novelty:** 3
**Q2(2) Significance/Impact:** 3
**Q2(3) Correctness/Technical Quality:** 3
**Q2(6) Clarity Of Writing:** 3
**Q6 Overall Score:** 7
**Q8 Confidence In Your Score:** 3

**Q1 Summary And Contributions:**

The paper addresses the problem of learning Bayesian networks from data. In particular, it is concerned with learning the posterior distribution over networks, or to be precise sampling from this posterior. A novel method for doing this is developed by adapting the recent GFlowNet method. The results show the technique to be competitive with current stat-of-the-art approaches.

**Q2 Assessment Of The Paper:**

More detailed information regarding each of these aspects is given below:

**Q2(4) Quality Of Experiments (Optional):**

3: Good: The experimental evaluation is adequate, and the results convincingly support the main claims.

**Q2(5) Reproducibility:**

3: Good: Key resources (e.g., proofs, code, data) are available and key details (e.g., proofs, experimental setup) are sufficiently well-described for competent researchers to confidently reproduce the main results.

**Q3 Main Strengths:**

The paper presents the technique clearly considering that it is inevitably quite mathematically complex and involves understanding GFlowNet before the contribution can be appreciated.

The results show that the technique is competitive with other existing methods. The exact quality depends on exactly what criterion one wishes to use, but the presented method is amongst the best on all 3 studied and doesn't appear ro be dominated by any one other technique.



**Q4 Main Weakness:**

I couldn't see any mention of the time the algorithm takes in comparison to other approaches. This is an important aspect in evaluating whether the technique is feasible for a real application. Indeed one could argue that the evaluation is only truly fair if all algorithms are allowed the same time budget.

The technique appears to work in the space of DAGs. In most situations, we would probably ideally want to sample from the space of probability distributions (i.e. the space of MECs). However, this is not a problem unique to this approach.

**Q5 Detailed Comments To The Authors:**

The paper would benefit from motivating the need for calculating a posterior in the introduction. It currently has a single sentence that basically says "this is what we want to do" without explaining why it is useful.

The second real-world experiment (section 6.4) is rather weak. It lacks a thorough comparison with different approaches and requires some modifications which are not fully explained. It doesn't really add much to the previous results which demonstrate the technique works well in a systematic synthetic setting and a real-world setting. If it is adding something to the paper I have failed to notice, it would be better to expand it to properly describe its contribution and setting. If it isn't adding much, it would be better to reclaim this space to expand other sections.

**Q7 Justification For Your Score:**

The paper presents an approach that evaluation suggests is at least as good as existing ones and appears to use a considerably different method.

**Q9 Complying With Reviewing Instructions:**

1: Yes.

---

### Decision · Program_Chairs · 2022-05-15

**Decision:**

Accept (Poster)

**Comment:**

Meta Review: The paper addresses the problem of learning Bayesian networks from data and presents a novel approach to sampling BNs from the posterior distribution given data. The experimental evaluation is reasonable and shows the proposed method is competitive with existing methods. The paper is generally well-written. Reviewers expressed concerns about the proposed approach's computational efficiency and time complexity, which should be discussed.